# Research on the Impact of Regular Exercise Behavior of College Students on Academic Stress and Sleep Quality during the COVID-19 Pandemic

**DOI:** 10.3390/healthcare10122534

**Published:** 2022-12-14

**Authors:** Ming-Zhu Yuan, Chao-Chien Chen, I-Shen Chen, Cheng-Chia Yang, Chin-Hsien Hsu

**Affiliations:** 1School of Education, Fuzhou University of International Studies and Trade, Fuzhou 350202, China; 2Department of Leisure and Recreation Management, Asia University, Taichung 413305, Taiwan; 3Department of Leisure Industry Management, National Chin-Yi University of Technology, Taichung 411030, Taiwan; 4Department of Healthcare Administration, Asia University, Taichung 413305, Taiwan

**Keywords:** COVID-19, regular exercise behavior, academic stress, sleep quality, student

## Abstract

When college students face the COVID-19 pandemic and new learning challenges simultaneously, how to reduce or alleviate their own academic stress has become a topic of concern to students and their parents. The psychological and physiological benefits of regular exercise have been confirmed by related studies. This study aimed to explore the impact of college students’ regular exercise behavior on academic stress and sleep quality during the COVID-19 pandemic. This study used a purposive sampling method to collect data through online questionnaires posted to relevant college student groups in northern, central, southern, and eastern Taiwan and the outlying islands. A total of 320 questionnaires were collected, with a response rate of 91.4%; based on 304 valid questionnaires. The validity rate was 95%. The obtained data were entered in SPSS 24.0 statistical software, and the correlation between variables was analyzed with AMOS 24.0 statistical software. The results show that hypothesis 1 is established, that is, regular exercise behavior of college students during the COVID-19 pandemic has a significant negative impact on academic stress, meaning that during the COVID-19 pandemic, if college students can use their spare time to make exercise part of their life, such a regular schedule will help reduce their academic stress. In addition, the empirical results show that hypothesis 2 is established, that is, regular exercise behavior of college students during the COVID-19 pandemic has a significant positive impact on sleep quality. A possible reason is that under the COVID-19 pandemic, the efficiency of the body to absorb oxygen is increased through regular exercise, which reduces pressure and improves sleep quality. Hypothesis 3 is also confirmed, that is, the academic stress of college students during the COVID-19 pandemic has a significant negative impact on sleep quality. The reason may be that many leisure and social activities have been suspended during the COVID-19 pandemic, and thus college students exercised and studied during the time they originally intended for leisure and social activities, which reduced their academic stress, stabilized their mood, and improved their sleep quality.

## 1. Introduction

Sports have long brought people joy in life and shaped the awareness of consumption and cultural identity of local communities. Ho [1] pointed out that the American education system attaches great importance to sports, regards sports as a link to education and uses sports to cultivate outstanding talents and leaders. Furthermore, ninety percent of corporate CEOs have been members of school teams during college. The coronavirus disease 2019 (COVID-19) outbreak in 2020 has affected people’s lives and exercise habits. The pathogen is caused by the 2019 coronavirus (2019 nCoV). COVID-19 is primarily transmitted through droplets and close contact. Upper respiratory tract symptoms are commonly observed.

With relevant research on epidemiology, pathogenesis, clinical manifestations, diagnoses, and treatments of COVID-19 [2], the relationship between exercise and viruses is gaining increasing attention.

In other words, the development of sports behavior is significant to individuals. According to the results of a survey on the current situation of sports by [3], the proportion of the population participating in sports in Taiwan reached 82.8%. A high sports participation rate of more than 80% has been maintained for 13 consecutive years, since 2008, and the proportion of the population that takes regular exercise has reached 33.0%. Since 2014, it has remained stable at 33.0% or more for 7 consecutive years, although there was a slight reduction in the proportion of the population that took regular exercise during 2020, which was due to the restrictions of the COVID-19 pandemic. It can be seen from these data that the concept of regular exercise habits has gradually been accepted by Chinese people, as well as the impact of the COVID-19 pandemic on the lives of the general public. A relevant study also highlighted the important role of exercise during the COVID-19 pandemic [4]. The study by [5] indicated that cardiopulmonary exercise testing (CPET) played an important role in clinical assessments of convalescent COVID-19 patients as well as research aimed at understanding the long-term health effects of SARS-CoV-2 infection.

While college students are just leaving the high/vocational school stage and are under the pressure of entering a higher education institution, they do have more free time to participate in leisure sports. This phenomenon is in line with the results presented by [6] regarding the participation in sports of students at all levels in the 2019 school year. The research results show that, in terms of the percentage of students at all levels of schools who consciously have a positive attitude towards and interest in sports, 67.7% of elementary school students tend to agree, 61.2% of junior high school students tend to agree, 61.9% of senior high/vocational school students tend to agree, and 67.7% of college students tend to agree. Although college students are more active in participating in leisure sports, they inevitably face the academic stress brought about by changes in the learning environment. This result echoes the findings of [7], which pointed out that schoolwork is the main source of stress for freshmen and sophomores. The academic stress of freshmen comes from adapting to a new style of learning. During the COVID-19 pandemic, schools have set pandemic prevention measures for classrooms in response to the government’s pandemic prevention regulations, and because the students are adapting to the teaching styles of teachers from their first to fourth year in college, it is always a source of academic stress. Academic stress is already the highest among sophomores; thus, when college students face the COVID-19 pandemic and new learning challenges simultaneously, how to reduce or alleviate their own academic stress has become a topic of concern to them and their parents. The psychological and physiological benefits of regular exercise have been confirmed by related studies [8,9], and regular exercise can improve immunity against COVID-19 infection. Therefore, during the COVID-19 pandemic, understanding the impact of regular exercise behavior on college students’ academic stress is one of the focuses of this study.

Sleep quality is a good indicator of healthy living. Tempesta et al. [10] found that sleep functions to regulate emotions. A lack of sleep can lead to increased negative and decreased positive emotions. In contrast, good sleep quality can improve immunity against COVID-19 infection. However, with the development of technology networks, the frequency of using 3C products by college students has also increased, and various factors, such as the colorful online world and participation in club activities, have often changed the sleep habits of college students. The lack of sleep among college students continues to gain more and more attention from researchers [11,12]. The results of a survey on students at all levels of schools in the 2019 school year by the Taiwan Sports Administration showed that, according to the conscious sleep status of students at all school levels, the proportion of elementary school students who were satisfied with their sleep status was 46.8%, of those dissatisfied 14.1%, of junior high school students that were satisfied and dissatisfied 29.6% and 23.7%, respectively, of satisfied and dissatisfied senior high/vocational school students 18.2% and 37.6%, of satisfied and dissatisfied college students 24.6% and 28.7%, and overall 28.2% and 27.2%, respectively. These results show that the higher the school level, the more dissatisfied the students were with their sleep conditions. In other words, sleep quality has gradually become another hidden worry and barrier for college students to construct good quality of life, and sleep quality is even more important during the COVID-19 pandemic. It has been empirically proven that regular exercise can improve sleep quality [13]. Therefore, exploring the impact of regular exercise behaviors on the sleep quality of college students during the COVID-19 pandemic is another focus of this study.

According to the literature review, research on regular exercise behaviors mostly focused on middle-aged and elderly people [14,15,16,17] and women or pregnant women [18,19,20]. Although a small part of research related to regular exercise is focused on students, few related discussions focused on the regular exercise of college students. In view of the trend of the declining birthrate, it is imperative for relevant units in colleges to understand the relevant issues that college students are currently facing to help them in their college life. It is urgent for college students to establish an understanding regarding the impact of regular exercise on their lives during the COVID-19 pandemic. This study’s objective was to further understand the current trend in college students’ regular behaviors and the impact of this trend on college students who want to balance heavy schoolwork with their leisure lives. The findings from this study could help to understand the impact of regular exercise on sleep quality through empirical research.

## 2. Literature Review

The concept of regular exercise has gradually been accepted by Chinese people in recent years, especially because most people are unable to exercise regularly due to various factors, such as schoolwork, work, and daily life, and the phrase “no time” has become the intuitive response of most people regarding participating in regular exercise. Yao [21] suggested that the general definition of regular exercise is when an individual continuously and regularly engages in physical activities. A more refined definition can be based on the type, frequency, duration, and intensity of exercise. The Sports Administration [22], defined different exercise behaviors more clearly by pointing out these are divided into five major stages if the exercise behavior is considered a change in history “from nothing to something.” In the first stage, the idea of participating in sports has not yet been established; in the second stage, the intention of exercise behavior begins to develop, but no actual behavior occurs; the third stage is the beginning of participation in sports, but the behavior is scattered and irregular; the fourth stage is the beginning of regular exercise behavior, but the duration is less than 6 months; the fifth stage is when the exercise behavior is regular, and the duration has exceeded 6 months. According to the abovementioned narrative of exercise behavior by the Sports Administration, we can see that the process of people participating in sports is a continuous one and varies according to the individual’s different levels of involvement. This perspective also agrees with [18], who indicated that participation in continuous exercise is a complex behavior determination mechanism. The decision is affected by interactions between environmental and personal factors. When college students enter different learning environments from high/vocational schools, their perceptions of time use and sports participation have also changed. According to Chao and Tseng [23], in terms of the process of human physiological and psychological development, schooldays are a critical period for shaping and fixing personal lifestyles, healthy behavior, and attitudes. Lee and Yu [24] more clearly stated that college students are in the stage of convergence between college sports and national exercise, and students at this stage are temporarily relieved of the pressure of entering a higher educational institution and can have more time to choose their favorite sports activities, which has considerable impact on the development of the entire sport. This argument is consistent with research by [25], who indicated that university is the best time to cultivate lifelong exercise habits, where 84.7% of college students with regular exercise habits still maintained the habit of regular exercise after graduation. On the contrary, 81.3% of college students without regular exercise habits remained at the same amount of physical activity after graduation or even exercised less. In view of this, it is important to understand the influence of regular exercise behavior of college students on themselves. Different studies have adopted different measurement scales for the measurement of regular exercise behaviors. Hong stated that [17]—based on the fact that a lack of physical exercise has become the fourth-largest risk factor for death in the world—in 2010, the World Health Organization revised the recommended amount of physical activity for a healthy adult to complete at least 150 min of moderate-intensity aerobic physical activity per week or at least 75 min of high-intensity aerobic physical activity per week. In 2011, the Health Promotion Administration also put forward the slogan of “Move for 150, be healthy,” and its content encourages adults over 18 years of age and the elderly to develop dynamic lifestyle habits and exercise for at least 10 min each time, in order to accumulate 150 min of moderately strenuous activity each week.

Different stages of schooling bring different levels of academic stress. Lin [26] stated that academic stress is a feeling of oppression formed after students’ subjective assessment of the requirements for academic performance according to personal factors, the external environment in the process of schoolwork, and their perception of the disturbing factors in the environment that hinder academic performance, which in turn triggers a chain reaction of physical, cognitive, emotional, and behavioral dimensions. This point of view suggests that the source of student academic stress is multifaceted, rather than a single cause. Lo et al. [27] further put forward that the high academic stress of students may also stem from self-expectation and parents’ intervention in their schoolwork and excessive expectations, which may cause greatly increased pressure. Competition from peers is another source of stress. In addition to the abovementioned factors in students’ academic stress, Ref. [28] further described it from the perspective of personal construction and pointed out that heavy academic stress may reflect learners’ perceptions of the external context, such as teachers’ and parents’ requirements for schoolwork or the curriculum design, but the stress may also be constructed by the students themselves during the learning process. However, academic stress is not completely useless at the learning stage: sometimes it is the motivation for students to move forward. As [29] suggested, high academic stress may be caused by learners’ perceptions of external contexts in the process of self-adjustment, such as parental expectations, schoolwork requirements, and curriculum design, or students’ personal active construction during the learning process. While appropriate pressure is the driving force for progress, too much or too little pressure still has a profound impact on students’ physiology, psychology, and social interpersonal relationships. In other words, it is a more important issue to understand the source of academic stress faced by college students and give them further appropriate assistance or solutions. Most researchers used the personal, family, college, and social dimensions for the measurement of academic stress [30].

Sleep is like a nutrient that humans need for survival, and it plays an important role in extending life. Wang et al. [31] pointed out that sleep quality plays an important role in personal health and life satisfaction, and is also a necessity for promoting physical and mental health. From this point of view, people’s attention to the issues of sleep quality is the awareness that it has a close relationship with overall health and quality of life. According to Chiu and Lee [32], sleep quality is an evaluation standard that measures a person’s satisfaction with their quality of sleep with a subjective qualitative and objective quantitative approach. In other words, through this evaluation standard, people can know the pros and cons of their quality of sleep. For example, Ref. [33] took 175 female college students as research subjects to explore the relationship between their physical activities and sleep quality, and found that the group with high physical activities had better overall sleep quality than the group with low physical activities. The benefits of good-quality sleep for the body and mind have been the focus of many studies. For example, Ref. [34] pointed out that sleep quality is very important for personal health, happiness, and efficiency. Cherng and Shiu [35] held a similar view, and considered that after adequate sleep, mental alertness increases, which can improve work performance. It can be seen from related literature that different researchers used different tools to measure sleep quality. Wang and Huang [36] stated that Taiwan’s most commonly used sleep quality scales include the Athens Insomnia Scale (AIS), Epworth Sleepiness Scale (ESS), and Pittsburgh Sleep Quality Index (PSQI).

Relevant studies have shown that regular exercise behavior has an impact on academic stress [37,38]. Regular exercise behavior has a positive effect on sleep quality [39,40,41,42,43], academic stress has an impact on sleep quality [44,45,46].

## 3. Research Method

### 3.1. Research Structure

This study aimed to explore the impact of regular exercise behavior of college students on academic stress and sleep quality. According to the research purpose and related literature, the proposed research structure is shown in Figure 1.

### 3.2. Research Hypotheses

This study proposes the following hypotheses. 

**Hypothesis** **1** **(H1).** 
*The impact of regular exercise behavior on academic stress is significant.*


**Hypothesis** **2** **(H2).** 
*The impact of regular exercise behavior on sleep quality is significant.*


**Hypothesis** **3** **(H3).** 
*The impact of academic stress on sleep quality is significant.*


### 3.3. Research Subjects

Relevant research methods indicate that samples of more than 200 are considered medium-sized. If structural equation model analysis results are required, a study should not include fewer than 200 [47]. When analyzing structural equation models, the sample number is recommended to be between 100 and 150 [48]. The current study recruited 350 college students during the influence of COVID-19 through purposive sampling to participate in a survey questionnaire. A total of 320 questionnaires were recovered, for a return rate of 91.4%. The number of valid questionnaires was 304, with an effective recovery rate of 95%. The authors of [49] indicate that the measures of composite reliability (CR) and average variance extracted (AVE) of the questionnaire should be used as the chi test of convergent validity. For a good convergent validity of a questionnaire, research recommends a CR value greater than 0.6 and an AVE value greater than 0.5 [50]. This study explored the impact of college students’ regular exercise behaviors on academic stress and sleep quality during the COVID-19 pandemic. The data were collected using web-based questionnaires through purposive sampling to include college students in Taiwan. The web-based questionnaire was created as a Google Form. The link was posted in related groups of college students in northern, central, southern, and eastern Taiwan along with outlying islands. The questionnaire distribution period was from 1 August to 31 August 2021. Among the valid samples, 112 respondents were male, and 192 were female. A majority of respondents were 18–20 years old, totaling 154 people. Most of the respondents lived in central Taiwan, totaling 157 people. Most participants had weekly petty cash amounts of NTD501–1000, totaling 82 people. Most of the participants attended public universities (179), and 167 were enrolled at science and technology universities.

### 3.4. Research Tools

The content of this research questionnaire was mainly compiled with reference to related documents and revised questionnaires of [51,52,53]. The questionnaire was divided into four parts, with a total of 54 questions. Two experts from the field of recreational sports and one expert from health management were consulted to evaluate the questionnaire and provide suggestions for final revision. A pretest questionnaire was conducted in mid-July 2021 based on the suggestions from experts regarding its necessary revisions. The pretest primarily targeted college students in Taiwan who were conducive to the revision of the questionnaire in terms of its appropriateness and readability to enhance its content validity. The questionnaire was divided into four parts with a total of 54 questions. This study used CFA to verify the reliability and validity of the questionnaire and referred to the modification indices (MIs) for items deleted [54]. The following comprise the questions that were deleted. From the regular exercise scale: 1. Regular exercise makes me physically strong. 3. Regular exercise makes people enjoy themselves. 5. Regular exercise boosts self-confidence. 9. Exercise boosts friendship. 10. Everyone is supposed to schedule regular time for exercise. 13. Exercise usually takes priority when scheduling daily plans. 14. Exercise cannot be missed, even in extremely busy times. From the academic stress scale: 5. I feel anxious due to the academic competition among my classmates. 6. I suffer violent mood swings that affect my study. 11. I feel anxious because I can’t understand what the teacher is explaining during class. 12. My study is interfered with by my classmates. 14. I feel pressured because my parents want me to enter the next level of education (a graduate institute). From the sleep quality scale: 1. I cannot fall asleep within 30 min. 3. I must get up halfway through my sleep to go to the bathroom. 4. I cannot breathe well when I sleep. 5. I snore or cough loudly when I sleep. 6. I feel cold when I sleep. 7. I feel hot when I sleep. 8. I have nightmares when I sleep. 9. I feel pain when I sleep. 11. In the past month, I have relied on medication to help me sleep. 12. In the past month, I have found it difficult to stay awake while riding or driving, eating, or engaging in daily social activities. Question 13—In the past month, I have been struggling to pull myself together to do what I am supposed to do—was deleted in this study.

### 3.5. Research Processing and Analysis

After the questionnaires were recovered, those that missed too many answers, had the same options checked for all questions, had the checked answers showing a repetitive pattern, or failed to answer according to the instructions were excluded. The valid questionnaires were counted, and invalid questionnaires were eliminated from the study results. The data were entered in IBM Statistical Package for the Social Sciences (SPSS) 24 statistical software, and the correlation between variables was analyzed with Analysis of Moment Structures Analysis of Moment Structures (AMOS) 24 statistical software.

SPSS is a statistical software program that performs analyses using a graphical user interface. This study used SPSS to conduct descriptive statistics to examine the subjects’ sample structure characteristics. The software sorted out and simplified the data and enabled researchers to understand the distribution of each item via frequency distribution and percentage.

AMOS statistical software has a visual graphical interface that can construct, modify, and analyze complex structural equation models. In this study, AMOS was used to create structural equation models and determine whether the variable path coefficient reached a significant level to verify the research hypotheses.

### 3.6. Ethical Considerations

The survey was conducted anonymously. We developed the questionnaire according to our review and analysis of the relevant literature and used content checking and reliability analysis to evaluate and refine the content of the questionnaire. The research assistant reconfirmed the respondents’ willingness to participate and have their data used in this study and reiterated that the data would be presented anonymously. Therefore, the design of our study and content of our manuscript were based on the principles of fairness, openness, and impartiality.

## 4. Research Results

### 4.1. Sample Characteristics

In this study, there were 304 valid samples, and the sample characteristics are shown in Table 1.

### 4.2. Measurement Model Analysis

(1)Verification of Convergent Validity

Bagozzi and Yi [49] pointed out that the convergent validity of questionnaire dimensions should be tested by composite reliability (CR) and average variance extracted (AVE), and suggested that the CR value of good convergent validity should be greater than 0.6 and the AVE value should be greater than 0.5 [50]. This study conducted convergent validity testing for the dimensions, such as regular exercise, academic stress, and sleep quality. The factor-loading values of all dimensions were between 0.75 and 0.95, the CR values were 0.89–0.94, and the AVE values were 0.64–0.80, consistent with the convergent validity standard recommended by [49,50,55]. Thus, according to the measured items, this research questionnaire has good convergent validity, as shown in Table 2, Table 3 and Table 4.

(2)Discriminant Validity

This study used the trust interval method (bootstrap) to test the discriminant validity of the questionnaire. First, in order to determine whether the questionnaire dimensions were completely correlated, the Pearson correlation coefficient trust interval between the questionnaire dimensions was examined, and the results show that the confidence interval of the questionnaire dimensions does not contain 1, which indicates that the discriminant validity of the questionnaire dimensions of this study is significantly good [56], as shown in Table 5 and Table 6.

(3)Structural Model Analysis

Referring to the recommendations of [55], this study used the seven indicators of the chi-square value (χ^2^), the ratio of χ^2^ to the degree of freedom, GFI, AGFI, RMSEA, CFI, and PCFI as the criteria to test the overall model fit. Bagozzi and Yi [49] pointed out that the smaller the ratio of χ^2^ to its degrees of freedom, the better, and the revised ratio in this study was 1.61; [55] indicated that the closer the GFI and AGFI values are to 1, the better, and the revised GFI and AGFI in this study were 0.90 and 0.88, respectively. Browne and Cudeck [57] considered that the best RMSEA value is less than 0.08, and the modified RMSEA value in this study was 0.04; the CFI best standard value is greater than 0.90, and the modified CFI in this study was 0.97; the PCFI needs to be at least greater than 0.50, and the revised PCFI of this study was 0.86. These results show that the overall fit index of this study reached the standard, as shown in Table 7.

(4)Research Results

Table 8 shows that the impact of regular exercise behaviors on academic stress was significant, and H1 was established. The impact of regular exercise behavior on sleep quality was significant, and H2 was established. The impact of academic stress on sleep quality was significant, and H3 was established.

(5)Discussion and Suggestions

i.Discussion

It can be seen from Figure 2 and Table 8 that H1 is established, that is, the regular exercise behavior of college students during the COVID-19 pandemic has a significant negative impact on academic stress. The results of this study are similar to those of [38], meaning that during the COVID-19 pandemic, if college students can use their spare time to make exercise part of their life, such a regular schedule will help reduce their academic stress. In addition, the empirical results show that H2 is established, that is, the regular exercise behavior of college students during the COVID-19 pandemic has a significant positive impact on sleep quality. This research result is consistent with [43], and a possible reason is that under the COVID-19 pandemic, the efficiency of the body to absorb oxygen is increased through regular exercise, which reduces pressure and improves sleep quality. H3 is also confirmed, that is, the academic stress of college students during the COVID-19 pandemic has a significant negative impact on sleep quality. The results of this study are similar to [46], and the reason may be that many leisure and social activities have been suspended during the COVID-19 pandemic; thus, college students exercise and study during the time they originally intended for leisure and social activities, which reduced their academic stress, stabilized their mood, and improved their sleep quality.

ii.Suggestions

(1)For college students

According to H2, the impact of regular exercise behavior on the sleep quality of college students during the COVID-19 pandemic has reached a positive significant level. Therefore, it is recommended that college students with no regular exercise habits can actively participate in college physical education courses and sports club courses to learn about different sports or popular leisure sports, in order to find and learn about the sport that suits their abilities and interests under the COVID-19 pandemic through the process of participation, which can further open opportunities to cultivate their personal exercise habits. For example, pickleball is currently promoted for college students. Compared to badminton and tennis, pickleball is suitable for college students who lack sports experience, as the exercise intensity can be high or low, the rules are simple, and it requires only basic movements. It is especially easy for college students who have been exposed to tennis, billiards, or badminton to learn. Exposure to different types of sports opportunities can construct regular exercise behaviors more often for college students during the COVID-19 pandemic, which can help release psychological and psychological pressure and improve their quality of sleep. Moreover, it is recommended that college students with regular exercise habits can increase their exercise time and exercise intensity during the COVID-19 pandemic to further improve their quality of sleep. *CommonWealth Magazine* [58] suggested that it is best to choose light- or medium-intensity exercises when exercising at night, as this level of activity can help people fall asleep faster and further improve sleep quality; however, it is better to avoid exercising within an hour before going to bed, in order to give the body enough time to relax. Therefore, under the COVID-19 pandemic, college students can use their spare time at night to engage in regular exercise of appropriate intensity that will not overstimulate the nervous system and make it easier to fall asleep, further allowing the body to get better rest. Of course, college students’ acceptance of 3C products is high at present; therefore, the relevant sports app (sports application) or online interactive fitness courses can help college students increase their interest in sports during the COVID-19 pandemic in an interesting environment, which will further cultivate their regular exercise habits.

(2)For Event College-Related Units

The results indicated that the regular exercise behavior of college students during the COVID-19 pandemic had a significantly negative impact on academic stress. Therefore, it is recommended that relevant college units arrange appropriate physical education and sports-related courses for college students during their study period, so they can develop the habit of regular exercise in a gradual and progressive manner. For example, swimming lessons are compulsory at some colleges and universities. Although the motive is to cultivate regular exercise habits, some students may be uncomfortable with swimming lessons and even reject the sport because of the pandemic, personal, or water-related factors. Therefore, during the COVID-19 pandemic, it is important to plan physical education curricula according to the students’ needs and motivations to build up their regular exercise habits. On the other hand, college administrators can promote sports-related experience activities for college students who do not have regular exercise habits so that they can balance their life and work by participating in sports and reducing their academic stress, e.g., courses that provide a basic understanding of high-altitude rope erection and operation, which can be connected to rock climbing or rope-walking related sports. Through the planning of physical education courses, students can spend their spare time training for sports that interest them, and then develop regular exercise habits. In addition, it is recommended that college management units encourage the diversified development of student sports clubs and vigorously support the college’s competitive sports teams, which will increase the importance attached to sports by the whole college and provide opportunities that stimulate students’ motivation to participate in sports, thus further promoting the cultivation of regular exercise habits. Moreover, it is recommended that related college units can regularly organize intramural athletic sports competitions for college students with sports habits, such as campus marathons and interdepartmental ball games. Via such activities, students can be encouraged to actively prepare for events, thereby enhancing the atmosphere of regular sports in the college and reducing academic stress.

This study also showed that academic stress had a negative impact on sleep quality. It is suggested that college authorities relieve students’ academic stress through leisure sports interventions during the COVID-19 pandemic. Specifically, this can be done to reduce the barriers to student participation in sports during the COVID-19 pandemic. For example, the provision of ventilated and safe sports venues and spaces (such as outdoor sports fields), sound pandemic prevention measures, and reasonable fees and opening hours can reduce the barriers for students to participate in sports during the COVID-19 pandemic and make them more willing to engage in regular sports. Once the habit of regular sports is developed, the effect of reducing academic stress can be achieved, which in turn improves the sleep quality of college students.

(3)For Future Research

This study verifies the causal relationship between regular exercise behaviors, academic stress, and sleep quality. Moreover, we found that during the COVID-19 pandemic, some college students still chose to exercise, despite the fact that various preventive measures and social distance regulations made it more difficult to achieve regular exercise. Based on our findings, it is evident that with related restriction measurements during the COVID-19 pandemic, the definition of regular exercise changed, meaning that college students chose exercises with lower intensity so that they could continue to exercise. Exercise intensity is a new perspective that has been seldom discussed in the previous literature. While this study found that some college students still have regular exercise habits during the COVID-19 pandemic, their motivation to engage in exercise is waning due to repetition. Therefore, it is recommended that related theories can be targeted for further research in the future. For example, the theory of planned behavior can be an entry point to study the regular sports participation of college students from three aspects: attitude, subjective norms, and perceived behavior control. In terms of attitude, understanding how college students treat regular exercise for weight control, health promotion, or a happy mood during the COVID-19 pandemic can help strengthen their belief in regular exercise behaviors. In addition, regarding subjective norms, due to the dependence of college students on their peers and important others, once a supportive environment for regular exercise is established in a college or family, it will help college students to become more involved in sports. Finally, in terms of perceived behavioral control, choosing suitable exercise intensity and the timely absorption of new knowledge about sports will help college students have a deeper understanding of regular exercise behaviors. As learned from the above, different theoretical foundations will help future related research to further explore the factors that affect college students’ regular exercise behaviors during the COVID-19 pandemic. In addition to recommending relevant theories for future research, exploring the preceding variables that affect the intention of regular exercise behaviors under the COVID-19 pandemic are also worthy of in-depth understanding, such as whether individual differences in self-esteem, emotion, or personality have an impact on regular exercise behavior.

## 5. Conclusions

As the relevant government departments continue to promote the national sports atmosphere, the concept of Chinese people’s spontaneous and regular exercise is gradually popularized. Appropriate regular exercise helps people maintain physical fitness and increase positive emotional feelings. When college students face a learning environment and technological environment different from the previous one, not only has the learning mode changed but also the pressure they face is different from the past. Therefore, whether college students’ regular exercise behavior can have an impact on their academic stress and sleep quality has become the focus of research. The following main conclusions have been obtained from the empirical analysis of this study:

H1 is established: The impact of regular exercise behavior on academic stress is significant.

H2 is established: The impact of regular exercise behavior on sleep quality is positively significant.

H3 is established: The impact of academic stress on sleep quality is negatively significant.

## Figures and Tables

**Figure 1 healthcare-10-02534-f001:**
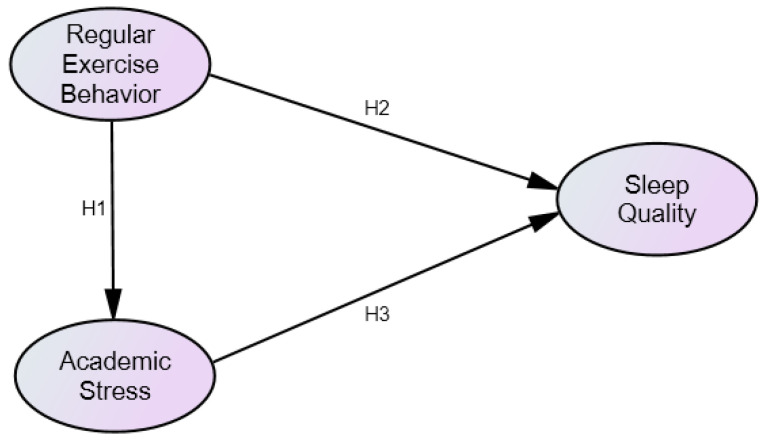
Research structure.

**Figure 2 healthcare-10-02534-f002:**
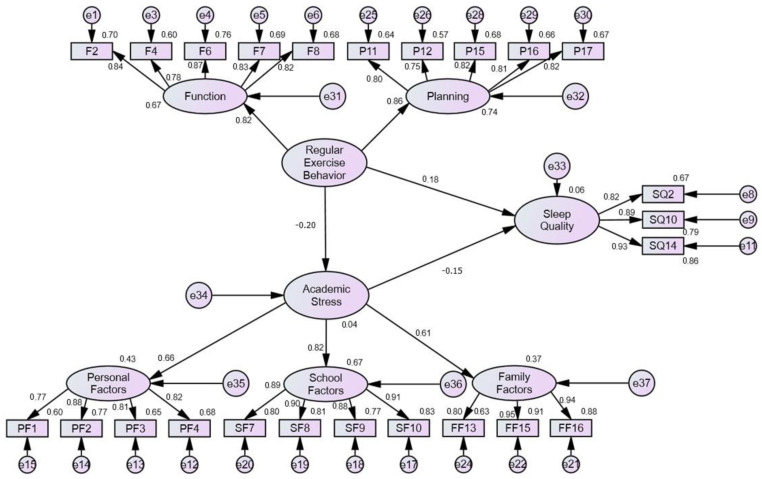
Model of impact of regular exercise behavior of college students on academic stress and sleep quality.

**Table 1 healthcare-10-02534-t001:** Sample characteristics.

Variable	Category	Number	Percentage
Gender	Male	112	36.8
	Female	192	63.2
Age	18–20 years old	154	50.7
	21–30 years old	121	39.8
	24–26 years old	13	4.3
	Over 27 years old	16	5.3
Grade	Freshman	29	9.5
	Sophomore	90	29.6
	Junior	64	21.1
	Senior	98	32.2
	Other	23	7.6
Average weekly expenses spent on leisure activities	North	98	32.2
	Central	157	51.6
	South	31	10.2
	East	12	3.9
	Outlaying island	6	2.0
Weekly petty cash	NTD 500 or lower	47	15.5
	NTD 501–1000	82	27.0
	NTD 1001–1500	62	20.4
	NTD 1501–2000	36	11.8
	NTD 2001 or more	77	25.3
University nature	Public university	179	58.9
	Private university	125	41.1
University education system	Science and technology university	167	54.9
	General university	134	45.1

**Table 2 healthcare-10-02534-t002:** Regular exercise behavior: confirmatory factor analysis.

Perspective	Index	Standardized Factor Loading	Nonstandardized Factor Loading	S.E.	C.R. (*t*-Value)	*p*	SMC	C.R.	AVE
Function	F2 Regular exercise improves my mental health.	0.84	1.00				0.70	0.91	0.68
	F4 Regular exercise helps people to be physically and mentally happy. It helps them forget their troubles.	0.78	0.89	0.06	15.63	***	0.60		
	F6 Regular exercise gives people a high sense of achievement.	0.87	1.08	0.06	18.78	***	0.76		
	F7 Regular exercise helps relieve stress.	0.83	0.94	0.05	17.17	***	0.69		
	F8 Regular exercise helps people maintain a good figure.	0.82	0.91	0.05	17.36	***	0.68		
Planning	P11 Failing to do regular exercise makes me feel uncomfortable.	0.80	1.00				0.64	0.89	0.64
	P12 I do leisure sports and activities with my family or friends.	0.75	0.89	0.06	14.14	***	0.57		
	P15 Everyone should develop an exercise plan.	0.82	1.01	0.06	15.70	***	0.68		
	P16 I take the initiative to collect information related to leisure sports and activities.	0.81	1.06	0.07	15.45	***	0.66		
	P17 I try to find time to do regular exercise (at least three times a week and 30 min each time).	0.82	1.10	0.07	15.47	***	0.67		

*** *p* < 0.001.

**Table 3 healthcare-10-02534-t003:** Academic stress—confirmatory factor analysis reached a significant level.

Perspective	Index	Standardized Factor Loading	Nonstandardized Factor Loading	S.E.	C.R. (*t*-Value)	*p*	SMC	C.R.	AVE
Personal factors	PF1 I feel uneasy if I cannot concentrate when preparing for an exam.	0.77	1.00				0.60	0.89	0.67
	PF2 I worry about not getting good results in my exams.	0.88	1.16	0.07	16.33	***	0.77		
	PF3 I feel very depressed when my academic performance declines.	0.81	1.03	0.07	14.47	***	0.65		
	PF4 I frequently worry about poor academic performance.	0.82	1.10	0.07	14.68	***	0.68		
School factors	SF7 I frequently feel the heavy amount of schoolwork is more than I can bear.	0.89	1.00				0.80	0.94	0.80
	SF8 I frequently feel very stressed because there are too many exams in school.	0.90	1.00	0.04	23.88	***	0.81		
	SF9 I feel very stressed because my teachers give too much homework.	0.88	1.02	0.05	22.46	***	0.77		
	SF10 I frequently feel drained because I must study so many subjects.	0.91	1.03	0.04	24.46	***	0.83		
family factors	FF13 I feel stressed when my parents often compare my accomplishments to the accomplishments of others.	0.80	1.00				0.63	0.92	0.80
	FF15 I feel very stressed because my parents have high expectations for my performance.	0.95	1.14	0.06	20.15	***	0.91		
	FF16 I feel very stressed because my parents expect me to be serious about studying.	0.94	1.12	0.06	19.61	***	0.88		

*** *p* < 0.001.

**Table 4 healthcare-10-02534-t004:** Sleep Quality—confirmatory factor analysis reached a significant level.

Perspective	Index	Standardized Factor Loading	Nonstandardized Factor Loading	S.E.	C.R. (*t*-Value)	*p*	SMC	C.R.	AVE
Sleep Quality	SQ2 I often wake up in the middle of the night or early morning.	0.81	1.00				0.66	0.90	0.77
	SQ10 I have had poor sleep quality in the past month.	0.89	1.07	0.06	18.40	***	0.79		
	SQ14 I would evaluate my overall sleep quality as poor over the past month.	0.93	1.15	0.06	18.91	***	0.86		

*** *p* < 0.001.

**Table 5 healthcare-10-02534-t005:** Regular exercise behavior—hindrance—bootstrap 95% confidence interval table of related coefficients.

Parameter			Estimate	Bias-Corrected		Percentile Method	
				Lower Boundary	Upper Boundary	Lower Boundary	Upper Boundary
Function	<-->	Planning	0.70	0.55	0.82	0.54	0.82

**Table 6 healthcare-10-02534-t006:** Academic stress—hindrance—bootstrap 95% confidence interval table of related coefficients.

Parameter			Estimate	Bias-Corrected		Percentile Method	
				Lower Boundary	Upper Boundary	Lower Boundary	Upper Boundary
Personal factors	<-->	School factors	0.55	0.45	0.65	0.44	0.64
Personal factors	<-->	Family factors	0.37	0.26	0.48	0.26	0.48
School factors	<-->	Family factors	0.51	0.39	0.62	0.39	0.62

**Table 7 healthcare-10-02534-t007:** Structural model.

Fitness Indicator	Acceptable Criteria	After Mode Revision	Model Fitness Judgment
χ^2^ (Chi-square)	The smaller the better	393.43	
Ratio of χ^2^	<3	1.61	Fit
GFI	>0.80	0.90	Fit
AGFI	>0.80	0.88	Fit
RMSEA	<0.08	0.04	Fit
CFI	>0.90	0.97	Fit
PCFI	>0.50	0.86	Fit

**Table 8 healthcare-10-02534-t008:** Summary of study hypotheses and validation results.

Hypotheses	Path	Path Coefficient	Remarks
1	Regular Exercise Behavior->Academic Stress	−0.20	accepted
2	Regular Exercise Behavior->Sleep Quality	0.18	accepted
3	Academic Stress->Sleep Quality	−0.15	accepted

## Data Availability

Not applicable.

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
