# Peer review of "Research on the Impact of Regular Exercise Behavior of College Students on Academic Stress and Sleep Quality during the COVID-19 Pandemic"

_healthcare, 2022, doi:10.3390/healthcare10122534_

Round 1
Reviewer 1 Report
Dear authors,
The topic “Research on the Impact of Regular Exercise Behavior of College 2 Students on Academic Stress and Sleep Quality during the 3 COVID-19 Pandemic” is an interesting and essential study. The manuscript provided meaningful findings for sedentary older hypertensive older people training. Here are several comments for the manuscript.
1. In the abstract, supplement the name of the analysis software used.
2. I suggest add student into keywords.
3. what is exactly the research question asked within this paper?
4. What about the reliability of sample? Could authors say more about that?
5. sample characteristics is too redundant, the author only needs a brief introduction, the content presented through the table is sufficient.
Author Response
Dear Editor,
Thank you very much for your patience.
For reviewer suggestions, answered in 1 files.
With the full text.
Best regards, Chih-Hung Tseng

Reviewer 2 Report
It's a please to review the manuscript titled "Research on the Impact of Regular Exercise Behavior of College 2 Students on Academic Stress and Sleep Quality during the 3 COVID-19 Pandemic" for "Healthcare" journal.
The paper is very interesting and pleasant to read. However, there some improvements that must be done in order to merit publication in this journal.
1. Please do not write your keywords in italics.
2. Please include a more general statement in the start of your abstract instead of stating directly by the aim
3. Lines 16 to 18, please try to rephrase in order to avoid redundance. I think it's better to use the past.
4. Lines 18 to 20, same remark, I think you could also include more results.
5. Line 28, please rephrase. Accoring to Ho, american education...etc. or Ho pointed out that...etc.
6. Line 33, please specify that it was by the sports administration [2]
7. Line 39, please write the full name "Coronavirus disease 2019 (COVID-19)" as it is the first time it appears in your study.
8. In my humble opinion you have to add a short background about the COVID-19 pandemic, approximately in line 41. please add a paragraph or even between 1-3 sentences writing about the pandemic.
There is also a lack of refrences concerning the pandemic, please add these papers:
* https://doi.org/10.1016/j.pcad.2021.04.005
* https://doi.org/10.3390/healthcare10071341
* https://doi.org/10.1080/17512433.2021.1902303
8. Line 66-67 same as line 28
9. Line 91 please delate extraparentheses
10. In general, please be careful and refer to template found in the instructions of authors in the journal website, there are a lot of mistakes, as an example, [11-14] instead of [11];[12];...etc.
11. Line 95-99 looks like copy/paste text. Please correct it.
12. Please describe correctely the objective of your study in the end of your introduction section.
13. Line 104-105, same remark as previously, please correct it in the whole document.
14. Line 119, please specify that it was conform to Wu. line 131, same thing with Sparling and Snow. Please do the same in the rest of the manuscript.
15. Line 254 please write the full name of SPSS and please cite it (IBM, addinsoft...etc.)
16. Same for AMOS
17. What were the exclusion criteria?
18. Table 1, I think you don't have to include accumulated percentage
19. Line 341, you don't have to specify "this study further discussed the results."
20. I am not really convicted by the research design, discussion section is really short, and suggestions were included in the conclusion section.
I think it's clearer and more logical to include suggestions / recommendations, future research direction in the discussion section and to keep a short conclusion, it's just my point of view.
21. Please specify what is sports APP
22. Line 446 to 453, fonts are different from the rest of the manuscript.
23. There is a lack of refrences in your study, please cite more articles including those that I suggested and other ones.
24. My last comment is concerning the English of the manuscript. There are a lot of grammatical mistakes and run-on sentences. Please revise carefully your manuscript and try to collaborate with an English native, a specialist or an English editing service.
Best wishes.
Author Response

(The authors gave the same response as above.)

Round 2
Reviewer 2 Report
Dear authors,
Thank you for this revision, the manuscript is now suitable for publication.
Good luck!